# FAST GEOMETRIC PROJECTIONS FOR
# LOCAL ROBUSTNESS CERTIFICATION

**Aymeric Fromherz**[*]
Carnegie Mellon University
Pittsburgh, PA, USA
fromherz@cmu.edu

**Klas Leino**[*]
Carnegie Mellon University
Pittsburgh, PA, USA
kleino@cs.cmu.edu

**Matt Fredrikson**
Carnegie Mellon University
Pittsburgh, PA, USA
mfredrik@cmu.edu

**Bryan Parno**
Carnegie Mellon University
Pittsburgh, PA, USA
parno@cmu.edu

**Corina Păsăreanu**
Carnegie Mellon University
and NASA Ames
Moffett Field, CA, USA
pcorina@cmu.edu

## ABSTRACT

Local robustness ensures that a model classifies all inputs within an $\ell_p$-ball consistently, which precludes various forms of adversarial inputs. In this paper, we present a fast procedure for checking local robustness in feed-forward neural networks with piecewise-linear activation functions. Such networks partition the input space into a set of convex polyhedral regions in which the network's behavior is linear; hence, a systematic search for decision boundaries within the regions around a given input is sufficient for assessing robustness. Crucially, we show how the regions around a point can be analyzed using simple geometric projections, thus admitting an efficient, highly-parallel GPU implementation that excels particularly for the $\ell_2$ norm, where previous work has been less effective. Empirically we find this approach to be far more precise than many approximate verification approaches, while at the same time performing multiple orders of magnitude faster than complete verifiers, and scaling to much deeper networks. An implementation of our proposed algorithm is available on GitHub[1].

## 1 INTRODUCTION

We consider the problem of verifying the *local robustness* of piecewise-linear neural networks for a given $\ell_p$ bound. Precisely, given a point, $x$, network, $F$, and norm bound, $\epsilon$, this entails determining whether Equation 1 holds.

$$\forall x'.\|x - x'\|_p \le \epsilon \implies F(x) = F(x') \tag{1}$$

This problem carries practical significance, as such networks have been extensively shown to be vulnerable to *adversarial examples* (Papernot et al., 2016; Szegedy et al., 2014), wherein small-norm perturbations are chosen to cause arbitrary misclassifications. Numerous solutions have been proposed to address variants of this problem. These can be roughly categorized into three groups: learning rules that aim for robustness on known training data (Croce et al., 2019; Madry et al., 2018; Wong & Kolter, 2018; Zhang et al., 2019; Xiao et al., 2019), post-processing methods that provide stochastic guarantees at inference time (Cohen et al., 2019; Lecuyer et al., 2018), and network verification (Balunovic et al., 2019; Cheng et al., 2017; Dutta et al., 2018; Ehlers, 2017; Fischetti & Jo, 2018; Gowal et al., 2018; Jordan et al., 2019; Katz et al., 2017; 2019; Singh et al., 2019b; Tjeng & Tedrake, 2017; Wang et al., 2018; Weng et al., 2018).

We focus on the problem of network verification—for a given model and input, determining if Equation 1 holds—particularly for the $\ell_2$ norm. Historically, the literature has primarily concentrated

---

[*]First two authors have equal contributions
[1]Code available at https://github.com/klasleino/fast-geometric-projections

on the $\ell_\infty$ norm, with relatively little work on the $\ell_2$ norm; indeed, many of the best-scaling verification tools do not even support verification with respect to the $\ell_2$ norm. Nonetheless, the $\ell_2$ norm remains important to consider for "imperceptible" adversarial examples (Rony et al., 2019). Furthermore, compared to the $\ell_\infty$ norm, efficient verification for the $\ell_2$ norm presents a particular challenge, as constraint-solving (commonly used in verification tools) in Euclidean space requires a non-linear objective function, and cannot make as effective use of interval-bound propagation.

Existing work on verifying local robustness for the $\ell_2$ norm falls into two primary categories: *(1)* expensive, but exact decision procedures, e.g., GeoCert (Jordan et al., 2019) and MIP (Tjeng & Tedrake, 2017), or *(2)* fast, but approximate techniques, e.g., FastLin/CROWN (Weng et al., 2018; Zhang et al., 2018). While approximate verification methods have shown promise in scaling to larger networks, they may introduce an additional penalty to robust accuracy by flagging non-adversarial points, thus limiting their application in practice. Exact methods impose no such penalty, but as they rely on expensive constraint-solving techniques, they often do not scale well to even networks with a few hundred neurons.

In this paper, we focus on bridging the gap between these two approaches. In particular, we present a verification technique for Equation 1 that neither relies on expensive constraint solving nor conservative over-approximation of the decision boundaries.

Our algorithm (Section 2) leverages simple projections, rather than constraint solving, to exhaustively search the model's decision boundaries around a point. The performance benefits of this approach are substantial, especially in the case of $\ell_2$ robustness, where constraint solving is particularly expensive while Euclidean projections can be efficiently computed using the dot product and accelerated on GPU hardware. However, our approach is also applicable to other norms, including $\ell_\infty$ (Section 3.3). Our algorithm is embarrassingly parallel, and straightforward to implement with facilities for batching that are available in many popular ML libraries. Additionally, we show how the algorithm can be easily modified to find certified lower bounds for $\epsilon$, rather than verifying a given fixed value (Section 2.3).

Because our algorithm relies exclusively on projections, it may encounter scenarios in which there is evidence to suggest non-robust behavior, but the network's exact boundaries cannot be conclusively determined without accounting for global constraints (Section 2, Figure 1b). In such cases, the algorithm will return unknown (though it would be possible to fall back on constraint solving). However, we prove that if the algorithm terminates with a robust decision, then the model satisfies Equation 1, and likewise if it returns not_robust, then an adversarial example exists (Theorem 1). Note that unlike prior work on approximate verification, our approach can often separate not_robust cases from unknown, providing a concrete adversarial example in the former. In this sense, the algorithm can be characterized as *sound* but *incomplete*, though our experiments show that in practice the algorithm typically comes to a decision.

We show that our implementation outperforms existing exact techniques (Jordan et al., 2019; Tjeng & Tedrake, 2017) by multiple orders of magnitude (Section 3.1, Table 2a and Section 3.3), while rarely being inconclusive on instances for which other techniques do not time out. Moreover, we find our approach enables *scaling to far deeper models* than prior work — a key step towards verification of networks that are used in practice. Additionally, on models that have been regularized for efficient verification (Croce et al., 2019; Xiao et al., 2019), our technique performs even faster, and scales to much larger models — including convolutional networks — than could be verified using similar techniques (Section 3.1, Table 2a). Finally, we compare our work to approximate verification methods (Section 3.2). We find that while our implementation is not as fast as previous work on efficient lower-bound computation for large models (Weng et al., 2018), our certified lower bounds are consistently tighter, and in some cases minimal (Section 3.2, Table 2b).

## 2 ALGORITHM

In this section we give a high-level overview of our proposed algorithm, and present some implementation heuristics that improve its performance (Section 2.2). We also propose a variant (Section 2.3) to compute certified lower bounds of the robustness radius. Correctness proofs for all of the algorithms discussed in this section are provided in Appendix A. Because our algorithm applies to arbitrary $\ell_p$

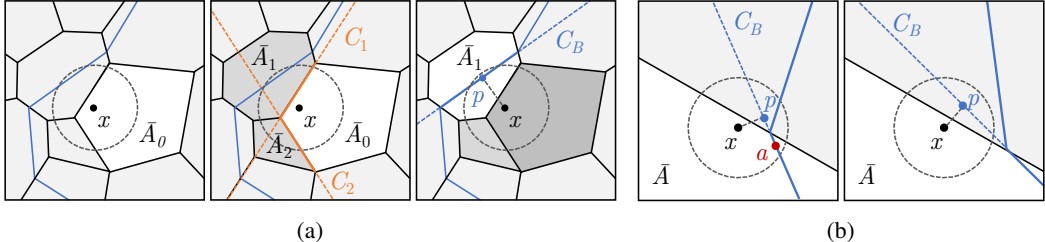

(a)                                                (b)

Figure 1: **(a)** Illustration of the FGP algorithm. We begin in $\bar{A}_0$ *(left)*. We see that activation constraints, $C_1$ and $C_2$ are in the $\epsilon$-ball, thus we enqueue $\bar{A}_1$ and $\bar{A}_2$ *(center)*. When searching $\bar{A}_1$, we see that a decision boundary, $C_B$, is within the $\epsilon$-ball. The projection $p$ onto $C_B$ is an adversarial example; we return not_robust *(right)*. **(b)** Illustration of the cases requiring FGP to return unknown when analyzing a boundary constraint, $C_B$, found within the $\epsilon$-ball about $x$. The true boundary is shown in solid blue; the infinite prolongation of the boundary we are projecting against is shown in dotted blue. In each case the projection, $p$, lies outside of $\bar{A}$, and is in fact not an adversarial example. In the case on the *left*, an adversarial example, $a$, exists in $\bar{A}$, while on the *right* local robustness is satisfied in $\bar{A}$. However, we cannot distinguish between these cases as $p \notin \bar{A}$ in both; in both cases we must return unknown.

norms, we use the un-subscripted notation $\| \cdot \|$ to refer to a general $\ell_p$ norm for the remainder of this section.

## 2.1    THE BASIC FAST GEOMETRIC PROJECTIONS ALGORITHM

We propose the *Fast Geometric Projections* (FGP) algorithm, which takes a model, $F$, an input, $x$, and a bound, $\epsilon$, and either proves that Equation 1 is satisfied (i.e., $\epsilon$-local robustness), finds an adversarial input at distance less than $\epsilon$ from $x$, or returns unknown. Our algorithm assumes $F(x) = \mathrm{argmax}\{f(x)\}$, where $f : \mathbb{R}^m \to \mathbb{R}^n$ is the function computed by a neural network composed of linear transformations with ReLU activations (i.e., $f$ is a feed-forward ReLU network).

The algorithm relies on an analysis of all possible *activation regions* around $x$. An activation region is a maximal set of inputs having the same *activation pattern* of ReLU nonlinearities.

Formally, let $f_u(x)$ denote the pre-activation value of neuron $u$ in network $F$ when evaluating $x$. We say that neuron $u$ is *activated* if $f_u(x) \geq 0$. An *activation pattern*, $A$, is a Boolean function over neurons that characterizes whether each neuron is activated. Then the *activation region*, $\bar{A}$, associated with pattern $A$ is the set of inputs that realize $A$: $\bar{A} := \{x \mid \forall u.(f_u(x) \geq 0) \iff A(u)\}$.

Because we assume that $F$ is a piecewise-linear composition of linear transformations and ReLU activations, we can associate the activation status $A(u)$ of any neuron with a closed half-space of $\mathbb{R}^m$ (Jordan et al., 2019). The *activation constraint*, $C_u$, for neuron $u$ and pattern $A$ is the linear inequality $w_u^T x + b_u \leq 0$, where $w_u$ and $b_u$ satisfy $\forall x \in \mathbb{R}^m.w_u^T x + b \leq 0 \iff A(u)$. The coefficients, $w_u$ are equal to the gradient of $f_u$ with respect to its inputs, evaluated at a point in $\bar{A}$. Crucially, because the gradient is the same at every point in $\bar{A}$, the constraints can be computed *from the activation pattern alone* via backpropagation. More details on this computation are given in Appendix B.

The intersection of these constraints yields the activation region $\bar{A}$, and the facets of $\bar{A}$ correspond to the non-redundant constraints. The convexity of activation regions follows from this observation, as does the fact that the *decision boundaries* are also linear constraints of the form $f(x)_i \geq f(x)_j$ for classes $i$ and $j$.

The FGP algorithm performs a search of all activation regions that might be at distance less than $\epsilon$ from $x$. We begin by analyzing the region, $\bar{A}_0$, associated with the activation pattern of the input, $x$, as follows.

First, we check to see if $\bar{A}_0$ contains a decision boundary, $C$, that is at distance less than $\epsilon$ from $x$; if so, we take the projection, $p$, from $x$ onto $C$. Intuitively, the projection is an input satisfying $C$ that is at minimal distance from $x$, i.e., $\forall x'.C(x') \implies \|x - p\| \leq \|x - x'\|$. Similarly we define the

distance, $d$, from $x$ to a constraint or decision boundary, $C$, as $d(x, C) := \min_{x':C(x')} \|x - x'\|$, i.e., $d$ is the distance from $x$ to its projection onto $C$. If $p$ does not have the same class as $x$ according to $F$ (or lies directly on the decision boundary between classes), then $p$ is an adversarial example. If $p$ has the same class as $x$, it means that the projection to the decision boundary is outside of the activation region we are currently analyzing; however, this is not sufficient to conclude that *no* point on the decision boundary is inside both the current activation region and the $\epsilon$-ball (see Figure 1b). Therefore, we return unknown.

Otherwise, we collect all activation constraints in $\bar{A}_0$; there is one such activation constraint per neuron $u$ in the network; thus each constraint corresponds to the neighboring region, $\bar{A}_0^u$, which has the same activation pattern as $\bar{A}_0$, except with neuron $u$ flipped, i.e., $A_0(u') \neq A_0^u(u') \iff u = u'$. For each constraint $C_u$, we check whether it might be at distance less than $\epsilon$ from $x$ using a geometric projection. Although $C_u$ is only valid in the polytope corresponding to the activation region $\bar{A}_0$, we compute the distance to the hyperplane corresponding to the infinite prolongation of $C_u$. As such, we only obtain a lower bound on the distance from $x$ to $C_u$. If the computed distance is smaller than $\epsilon$, we enqueue the corresponding activation region if it has not been searched yet. Thus the queue contains unexplored regions that might be at distance less than $\epsilon$ from $x$. The algorithm continues to analyze each enqueued region in the same way, each explored region enqueuing neighbouring regions that might be at distance less than $\epsilon$ from $x$, until the queue is empty. Exhausting the queue means that we did not find any adversarial examples in any activation region that might intersect with the $\epsilon$-ball centered at $x$. We therefore conclude that $x$ is locally $\epsilon$-robust. An illustration of a simple execution of this algorithm is shown in Figure 1a.

**Conditions on Adversarial Examples.** It is common in practice to include domain-specific conditions specifying which inputs are considered valid adversarial examples. For example, in the image domain, adversarial examples are typically subject to *box constraints* requiring each pixel to be in the range $[0, 1]$. These constraints can easily be incorporated into the FGP algorithm by checking them when we check if $F(x) \neq F(p)$, when a decision boundary is found. If $p$ is not a valid adversarial example, the algorithm would return unknown.

**Correctness.** We show that when the FGP algorithm returns not_robust, there exists an adversarial example, and when it returns robust, the model is locally robust at $x$, with radius $\epsilon$. However, the algorithm may also return unknown, in which case we do not claim anything about the robustness of the model. A complete proof of Theorem 1 is provided in Appendix A.1.

**Theorem 1.** *(1) When the FGP algorithm returns* not_robust, *there exists an adversarial example, $p$, such that $\|x - p\| \leq \epsilon$ and $F(x) \neq F(p)$. (2) When it returns* robust, *Equation 1 holds.*

## 2.2 HEURISTICS FOR FGP

To improve the performance and precision of our algorithm, we use several additional heuristics.

**Batching.** The gradient backpropagations for computing constraints and the dot product projections for calculating distances lend themselves well to batching on the GPU. In our implementation we dequeue multiple elements from the queue at once and calculate the constraints and constraint distances for the corresponding regions in parallel, leading to a speedup of up to 2x on large examples for a well-chosen batch size ($\sim$10-100).

**Caching Intermediate Computations.** We observe that activation constraints for the first layer are invariant across all regions, as there is no earlier neuron to depend on. Thus, we can precompute all such constraints once for all inputs to be analyzed.

**Exploring the Full Queue.** If the analysis of a decision boundary is inconclusive, the FGP algorithm, as presented, stops and returns unknown. We also implement a variant where we instead record the presence of an ambiguous decision boundary but continue exploring other regions in the queue. If we empty the queue after unsuccessfully analyzing a decision boundary, we nevertheless return unknown. However, oftentimes a true adversarial example is found during the continued search, allowing us to return not_robust conclusively.

Table 1 displays the results of experiments evaluating the impact of exploring the entire search queue instead of stopping at the first decision boundary (as described in Section 2.1). The experimental results show that this heuristic decreases the number of unknown results by approximately $50\%$ while

|  | Vanilla | | Full Queue | |
| Model | Time (s) | U | Time (s) | U |
| --- | --- | --- | --- | --- |
| mnist20x3 | 0.012 | 7 | 0.012 | 4 |
| mnist20x6 | 0.177 | 8 | 0.174 | 5 |
| mnist20x9 | 4.897 | 16 | 7.389 | 3 |
| mnist40x3 | 5.417 | 9 | 6.712 | 3 |

Table 1: Comparison of FGP and its variant with full exploration of the queue

only having a minor impact on the speed of the execution. Moreover, we reduce the number of unknown results to the extent that we recover the results obtained by GeoCert in every instance for which GeoCert terminates, except 3 non-robust points on mnist40x3, while nevertheless performing our analysis two to three orders of magnitude more quickly.

## 2.3 CERTIFIED LOWER BOUNDS

We now consider the related problem of finding a *lower bound* on the local robustness of the network. A variant of the FGP algorithm can provide certified lower bounds by using a priority queue; constraints are enqueued with priority corresponding to their distance from $x$, such that the closest constraint is always at the front of the queue. We keep track of the current certified lower bound in a variable, $\beta$. At each iteration, we set $\beta$ to the maximum of the old $\beta$ and the distance from $x$ to the constraint at the front of the queue.

The algorithm terminates either when all constraints at distance less than the initially specified $\epsilon$ were handled, or when a decision boundary for which the initial algorithm returns unknown or not_robust is found. In the first case, we return $\epsilon$; in the second, we return the value stored in $\beta$ at this iteration.

The proof of this variant is similar to the proof of the FGP algorithm. It relies on the following loop invariant, which we prove in Appendix A.2.

**Loop Invariant 1.** *(1) All activation regions at distance less than $\beta$ from $x$ were previously visited, (2) $\beta$ is always smaller than $\epsilon$, and (3) there is no adversarial point at distance less than $\beta$.*

## 3 EVALUATION

In this section, we evaluate the performance of our implementation of the FGP algorithm and its variant for computing lower bounds. Section 3.1 discusses the performance of FGP compared to existing tools that perform exact $\ell_2$ robustness verification. Section 3.2 compares the lower bounds certified by our FGP variant from Section 2.3 to those of other approximate certification methods. In short, we find that our approach outperforms existing exact verification tools by several orders of magnitude (2-4), and produces more precise lower bounds (3-25 times larger) than the relevant work on approximate certification.

Additionally, in Section 3.1, we explore the scalability of our algorithm. We find that when the networks are strongly regularized for verifiability, our approach even scales to CNNs. Finally, we remark on the flexibility of FGP with respect to the norm in Section 3.3, and observe that our approach is also faster than existing complete verification tools when performing $\ell_\infty$ robustness verification.

We performed experiments on three standard datasets: MNIST, Fashion-MNIST, and CIFAR10. We evaluated both on models trained for robustness using adversarial training (Madry et al., 2018), and on models trained for verifiability *and* robustness using maximum margin regularization (MMR) (Croce et al., 2019) or ReLU Stability (RS) (Xiao et al., 2019). We refer to each model by the dataset it was trained on, followed by the architecture of the hidden layers. For example, "mnist20x3" refers to a model trained on MNIST with 3 hidden layers of 20 neurons each. The "cnn" architecture refers to a common CNN architecture used to benchmark CNN verification in the literature; details are given in Appendix C. Models marked with "*" were trained with MMR; models marked with "†" were trained with RS; all other models were trained using PGD adversarial training (Madry et al., 2018). The hyperparameters used for training are given in Appendix C.

| | FGP | | | | | | GeoCert | | | | | MIP | | | | |
|---|---|---|---|---|---|---|---|---|---|---|---|---|---|---|---|---|
| Model | Time (s) | R | NR | U | TO | VRA | Time (s) | R | NR | TO | VRA | Time (s) | R | NR | TO | VRA | VRA$_{UB}$ |
| mnist20x3 | 0.012 | 87 | 8 | 4 | 1 | 0.82 | 10.99 | 82 | 9 | 9 | 0.77 | 97.02 | 58 | 9 | 33 | 0.54 | 0.84 |
| mnist20x6 | 0.174 | 79 | 9 | 5 | 7 | 0.74 | 78.93 | 49 | 9 | 42 | 0.44 | >120 | 0 | 0 | 100 | 0.00 | 0.85 |
| mnist20x9 | 7.389 | 44 | 16 | 3 | 37 | 0.42 | >120 | 10 | 10 | 80 | 0.10 | >120 | 0 | 0 | 100 | 0.00 | 0.82 |
| mnist40x3 | 6.712 | 54 | 6 | 3 | 37 | 0.51 | >120 | 16 | 11 | 73 | 0.16 | >120 | 0 | 0 | 100 | 0.00 | 0.89 |
| fmnist200x4* | 0.025 | 81 | 14 | 4 | 1 | 0.73 | 54.85 | 45 | 14 | 41 | 0.41 | >120 | 0 | 0 | 100 | 0.00 | 0.76 |
| fmnist200x6* | 0.087 | 66 | 12 | 3 | 19 | 0.60 | >120 | 28 | 5 | 67 | 0.26 | >120 | 0 | 0 | 100 | 0.00 | 0.75 |
| fmnist100x20* | 0.057 | 86 | 7 | 7 | 0 | 0.61 | >120 | 42 | 8 | 50 | 0.37 | >120 | 0 | 0 | 100 | 0.00 | 0.66 |
| cifar-cnn† | 0.058 | 86 | 14 | 0 | 0 | 0.27 | not supported | | | | | >120 | 0 | 0 | 100 | 0.00 | 0.27 |

(a)

| Model | FGP$_{LB}$ Mean Bound | FastLin Mean Bound | Median Ratio |
|---|---|---|---|
| fmnist500x4 | 0.124 | 0.078 | 0.329 |
| fmnist500x5 | 0.134 | 0.092 | 0.693 |
| fmnist1000x3 | 0.083 | 0.021 | 0.035 |

(b)

Table 2: **(a)** Comparison of $\ell_2$ local robustness certification (FGP vs. GeoCert vs. MIP) on 100 arbitrary test instances, including the median runtime, the certification result — either robust (R), not_robust (NR), unknown (U), or a timeout (TO) — and the corresponding Verified robust accuracy (VRA). The upper bound on the VRA (VRA$_{UB}$) is also given. Each instance is given a time budget of 120 seconds. Results are for $\epsilon = 0.25$, except on CIFAR10, where we use $\epsilon = 0.1$, as was used by (Croce et al., 2019). **(b)** Comparison of the lower bound obtained by FGP (Section 2.3) to that obtained by FastLin.

In each experiment, measurements are obtained by evaluating on 100 arbitrary instances from the test set. All experiments were run on a 4.2GHz Intel Core i7-7700K with 32 GB of RAM, and a Tesla K80 GPU with 12 GB of RAM.

## 3.1 LOCAL ROBUSTNESS CERTIFICATION

We first compare the efficiency of our implementation to that of other tools certifying local robustness. GeoCert (Jordan et al., 2019) and MIP (Tjeng & Tedrake, 2017) are the tools most comparable to ours as they are able to exactly check for robustness with respect to the $\ell_2$ norm. We used commit hash 8730aba for GeoCert, and v0.2.1 for MIP. Specifically, we compare the median run time over the analyses of each of the selected 100 instances, the number of examples on which each tool terminates, the result of the analyses that terminate, and the corresponding verified robust accuracy (VRA), i.e., the fraction of points that were both correctly classified by the model and verified as robust. In addition, we report an upper bound on the VRA for each model, obtained by running PGD attacks (hyperparameters included in Appendix C) on the correctly-classified points on which every method either timed out or reported unknown. Results are given for an $\ell_2$ norm bound of $\epsilon = 0.25$, with a computation budget of two minutes per instance. The results for these experiments are presented in Table 2a.

We observe that FGP always outperforms GeoCert by two to three orders of magnitude, without sacrificing precision; i.e., we rarely return unknown when GeoCert terminates. MIP frequently times out with a time budget of 120 seconds, indicating that we are faster by at least four orders of magnitude. This is consistent with the 100 to 1000 seconds per solve on a MNIST model with three hidden layers of 20 neurons each reported by Tjeng & Tedrake (2017); their technique performs best on the $\ell_1$ or $\ell_\infty$ norms. In addition, we find that FGP consistently verifies the highest fraction of points, and yields the best VRA. Moreover, on mnist20x3, FGP comes within 2 percentage points of the VRA upper bound. This suggests that FGP comes close to verifying *every* robust instance on this model.

**Models Trained for Verifiability.** Adversarial training (Madry et al., 2018), used to train the models in Sections 3.1 and 3.2, attempts to ensure that points on the data manifold will be far from the model's decision boundary. Even if it achieves this objective, it may nonetheless be difficult to verify because the performance of FGP depends not only on the decision boundaries, but also on the internal activation constraints. Thus, we expect certification will be efficient only when the points to be certified are also far from internal boundaries, leading to fewer regions to explore. Recent work has sought to develop training procedures that not only encourage robustness, but that also optimize

for efficient verification (Croce et al., 2019; Wong & Kolter, 2018; Xiao et al., 2019). Maximum margin regularization (MMR) (Croce et al., 2019) and ReLU stability (RS) (Xiao et al., 2019) are particularly suited to our work, as they follow from the same intuition highlighted above.

Using MMR, FGP is able to scale to much larger models with hundreds or thousands of neurons, and tens of layers, as shown in the bottom half of Table 2a. Here again, we see that FGP outperforms the other approaches in terms of both time and successfully-verified points. By comparison, while GeoCert also experienced improved performance on the MMR-trained models, our method remains over three orders of magnitude faster. MIP continued to time out in nearly every case.

We found that we were able provide even stronger regularization with RS, allowing us to scale to CNNs (Table 2a), which have far more internal neurons than even large dense networks. We found that these highly regularized CNNs verified more quickly than some of the less-regularized dense networks, though, as with other methods that produce verifiable CNNs, this came at the cost of a penalty on the model's accuracy.

## 3.2 CERTIFIED LOWER BOUNDS

We now evaluate the variant of our algorithm computing certified lower bounds on the robustness radius (Section 2.3). To this end, we compare the performance of our approach to FastLin (Weng et al., 2018), which is designed to provide quick, but potentially loose lower bounds on the local robustness of large ReLU networks. We compare the certified lower bound reported by our implementation of FGP after 60 seconds of computation (on models large enough such that FGP rarely terminates in 60 seconds) to the lower bound reported by FastLin; FastLin always reported results in less than two seconds on the models analyzed. The results are presented in Table 2b. The mean lower bound is reported for both methods, and we observe that on the models tested, FGP is able to find a better lower bound on average, though it requires considerably more computation time.

Because the optimal bound may vary between instances, we also report the median ratio of the lower bounds obtained by the two methods on each individual instance. Here we see that FastLin may indeed be quite loose, as on a typical instance it achieves as low as 4% and only as high as 69% of the bound obtained by FGP. Finally, we note that when FGP terminates by finding a decision boundary, if the projection onto that boundary is a true adversarial example, then *the lower bound is tight*. In our experiments, there were few such examples — three on fmnist500x4 and one on fmnist1000x3 — however, in these cases, the lower bound obtained by FastLin was very loose, achieving 4-15% of the optimal bound on fmnist500x4, and only 0.8% of the optimal bound on fmnist1000x3. This suggests that while FastLin has been demonstrated to scale to large networks, one must be careful with its application, as there may be cases in which the bound it provides is too conservative.

## 3.3 GENERALIZATION TO OTHER NORMS

The only aspect of FGP that depends on the norm is the projection and the projected distance computation, making our approach highly modular with respect to the norm. As such, we added support for $\ell_\infty$ robustness certification by providing a projection and a projected distance computation in FGP.

In this section, we compare the efficiency of this implementation to GeoCert (Jordan et al., 2019) and ERAN (Singh et al., 2019a). ERAN uses abstract interpretation to analyze networks using conservative over-approximations. As such, the analysis can certify that an input is robust, but can yield false positives when flagging an input as not robust. In this evaluation, we use the *DeepPoly* abstract domain, and the complete mode of ERAN which falls back on constraint solving when an input is classified as not robust.

Our experimental setup is similar to the one presented in Section 3.1. For each network, we arbitrarily pick 100 inputs from the test set and report the median analysis time of each tool. We give each tool a time budget of 120 seconds per input. We report the number of instances on which each tool terminates within this time frame, as well as the results of the analyses that terminate. Results are given in Table 3 for an $\ell_\infty$ norm bound of $\epsilon = 0.01$, which contains roughly the same volume as the $\ell_2$ ball used in most of our evaluation.

| | FGP | | | | ERAN+MIP | | | | GeoCert | | | |
|---|---|---|---|---|---|---|---|---|---|---|---|---|
| Model | Time (s) | R | NR | U | TO | Time (s) | R | NR | TO | Time (s) | R | NR | TO |
| mnist20x3 | 0.004 | 94 | 0 | 6 | 0 | 0.017 | 95 | 5 | 0 | 0.360 | 95 | 5 | 0 |
| mnist20x6 | 0.016 | 95 | 0 | 4 | 1 | 0.054 | 97 | 3 | 0 | 0.988 | 97 | 3 | 0 |
| mnist20x9 | 0.021 | 84 | 4 | 8 | 4 | 0.080 | 92 | 8 | 0 | 2.799 | 89 | 8 | 3 |
| mnist40x3 | 0.094 | 91 | 2 | 5 | 2 | 0.040 | 96 | 4 | 0 | 2.843 | 94 | 4 | 2 |
| fmnist100x6* | 0.012 | 85 | 7 | 8 | 0 | 0.982 | 91 | 9 | 0 | 0.750 | 91 | 9 | 0 |
| fmnist100x10* | 0.021 | 92 | 1 | 7 | 0 | 2.104 | 94 | 6 | 0 | 1.308 | 93 | 6 | 1 |

Table 3: Comparison of $\ell_\infty$ local robustness certification (FGP vs. ERAN vs. GeoCert) on 100 arbitrary test instances with a time budget of 120 seconds, including the median runtime and the certification result: either robust (R), not_robust (NR), unknown (U), or a timeout (TO). Results are for $\epsilon = 0.01$.

We refer to each model by the dataset it was trained on, followed by the architecture of the hidden layers. For example, "mnist20x3" refers to a model trained on MNIST with 3 hidden layers of 20 neurons each. Models marked with an asterisk were trained with MMR (Croce et al., 2019); other models were trained using PGD adversarial training (Madry et al., 2018).

We observe that we consistently outperform GeoCert by about two orders of magnitude, both on models trained with MMR and using adversarial training. On models trained for robustness (upper rows of Table 3), FGP is 0.5x to 5x faster than ERAN. In particular, FGP seems to scale better to deeper networks, while ERAN performs better on wider networks. Interestingly, ERAN does not benefit as much from MMR training as FGP and GeoCert. On large models trained with MMR, our tool is one to two orders of magnitude faster than ERAN, and ERAN is only about 2x faster than GeoCert.

Finally, while FGP is able to determine robustness for almost all inputs deemed robust by GeoCert or ERAN, it struggles to find adversarial examples for non-robust points, which results in a higher number of unknown cases compared to the $\ell_2$ variant. This is not highly surprising, as projections are better-suited to the Euclidean space than to $\ell_\infty$ space. While the projection in Euclidean space is unique, this is not always the case in $\ell_\infty$ space; we pick one arbitrary projection which thus might not lead to an adversarial example.

## 4 RELATED WORK

Our work can be grouped with approaches for verifying neural networks that aim to check local robustness exactly (Jordan et al., 2019; Katz et al., 2017; 2019; Tjeng & Tedrake, 2017); the primary difference is that our approach avoids expensive constraint solving at the price of incompleteness.

GeoCert (Jordan et al., 2019) is the closest work to ours; it aims to exactly compute local robustness of deep neural networks for convex norms. Unlike our approach, GeoCert computes the largest $\ell_p$ ball centered at an input point within which the network is robust. Our experimental comparison with GeoCert shows that our approach scales much better. This is not surprising as GeoCert relies on projections to polytopes, which are solved by a quadratic program (QP) with linear constraints. This enables to compute exactly the distance from a point to an activation constraint, instead of an underapproximation as our approach does. In contrast, our approach uses projections to affine subspaces, which have a simpler, closed-form solution. Lim et al. (2020) builds on GeoCert, but reduces the number of activation regions to consider by exploring the network in a hierarchical manner. Although Lim et al. reduce the number of QPs required for verification, their variant still relies on constraint solving. While they improve on GeoCert by up to a single order of magnitude, our approach consistently outperforms GeoCert by 2-4 orders of magnitude. MIP (Tjeng & Tedrake, 2017) is an alternative to GeoCert based on mixed-integer programming; it also requires solving QPs for the $\ell_2$ norm. We could fall back on similar techniques to provide a slower, but complete variant of our algorithm when our projections cannot reach a conclusion about a decision boundary.

Reluplex (Katz et al., 2017) and its successor, Marabou (Katz et al., 2019) are complete verification tools based on SMT solving techniques. Unlike our approach, Reluplex and Marabou do not support the $\ell_2$ norm. AI2 (Gehr et al., 2018) and its successor, ERAN (Singh et al., 2019b), are based on abstract interpretation (Cousot & Cousot, 1977), using conservative over-approximations to perform

their analysis, which leads to false positives, i.e. robust inputs incorrectly classified as not robust. A mode of ERAN enables complete verification by falling back on a constraint solver when an input is classified as not robust; however this tool does not support the $\ell_2$ norm either.

FastLin (Weng et al., 2018) exploits the special structure of ReLU networks to efficiently compute lower bounds on minimal adversarial distortions. CROWN (Zhang et al., 2018) later expanded this to general activation functions. Although FastLin has been shown to be very scalable, our experiments indicate that the computed bounds may be imprecise.

Recently, a quite different approach has been proposed for robustness certification. Randomized Smoothing (Cohen et al., 2019; Lecuyer et al., 2018) is a post-processing technique that provides a stochastic robustness guarantee at inference time. This approach differs from our approach in that it *(1)* modifies the predictions of the original model (increasing the complexity of making predictions), and *(2)* provides a *probabilistic* robustness guarantee that is quantified via a confidence bound. As such it provides an alternative set of costs and benefits as compared to static verification approaches. Its complexity also differs from that of FGP, as it is dependent primarily on the number of samples required to perform its post-processing of the model's output. We find that in our experimental setup, achieving the same probabilistic guarantee as the experiments described in (Cohen et al., 2019) requires $10^5$ samples, taking approximately 4.5 seconds per instance. Thus, for the models in our evaluation, FGP is on average faster or comparable in performance.

## 5 CONCLUSION

In this paper, we presented a novel approach for verifying the local robustness of networks with piecewise linear activation functions, that relies neither on constraint solving nor conservative over-approximations, but rather on geometric projections. While most existing tools focus on the $\ell_1$ and $\ell_\infty$ norms, we provide an efficient, highly parallel implementation to certify $\ell_2$-robustness. Our implementation outperforms existing exact tools by multiple orders of magnitude, while empirically maintaining the same or better precision under a time constraint. Additionally, we show that our approach is particularly suited to scale up network verification to *deeper* networks—a promising step towards verifying large, state-of-the-art models.

**Acknowledgments.** The work described in this paper has been supported by the Software Engineering Institute under its FFRDC Contract No. FA8702-15-D-0002 with the U.S. Department of Defense, Bosch Corporation, an NVIDIA GPU grant, NSF Award CNS-1801391, DARPA GARD Contract HR00112020006, a Google Faculty Fellowship, and the Alfred P. Sloan Foundation.

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

# A  Correctness Proofs

## A.1  Proof of Theorem 1

We show that when FGP returns not_robust, there exists an adversarial example, and when it returns robust, the model is locally robust at $x$, with radius $\epsilon$. However, the algorithm may also return unknown, in which case we do not claim anything about the robustness of the model.

*Proof.* In the first case, where FGP returns not_robust, the proof of Theorem 1 is trivial: we exhibit a point, $p$, such that $\|x - p\| \leq \epsilon$, and for which $F(x) \neq F(p)$.

The interesting case is when FGP returns robust. We prove by contradiction that in this case, $x$ is in fact locally robust with radius $\epsilon$.

Let us assume for the sake of contradiction that FGP returns robust, but there exists a point, $p$, such that $\|x - p\| \leq \epsilon$ and $F(x) \neq F(p)$. Let $\bar{A}_x$ and $\bar{A}_p$ be the activation regions associated with $x$ and $p$ respectively.

We define a *path* of activation regions as a sequence, $\bar{A}_0, \ldots, \bar{A}_k$, of activation regions such that the underlying activation patterns $A_i$ and $A_{i+1}$ differ in exactly one neuron for all $i$, and there exists at least one input, $x_i$, that has activation pattern $A_i$ for all $i$. For instance, in a network with three neurons, if $A_0 = (true, true, false)$, $A_1 = (true, false, false)$, and $A_2 = (true, false, true)$, and there exist inputs, $x_0$, $x_1$, and $x_2$ with activation patterns $A_0$, $A_1$, and $A_2$, then $\langle \bar{A}_0, \bar{A}_1, \bar{A}_2 \rangle$ is a path.

Our proof relies on three facts, that we prove hereafter:

1. There exists a path, $P$, from $\bar{A}_x$ to $\bar{A}_p$ where each region in the path contains at least one input at distance less than $\epsilon$ from $x$, and either $\bar{A}_x = \bar{A}_p$ or all $\bar{A}_i, \bar{A}_j$ in the path are different.

2. Our algorithm visits all regions in the path, $P$.

3. If a visited activation region contains an adversarial input, our algorithm either detects it, returning not_robust, or returns unknown.

Together, *(1)*, *(2)*, and *(3)* imply that if an adversarial point, $p$, exists, it resides in an activation region that would have been checked by FGP, which would have resulted in the algorithm returning not_robust or unknown, contradicting the assumption that it returned robust.

**(1) Existence of a Path**    Consider the segment going from $x$ to $p$ in a straight line. As $\|x - p\| \leq \epsilon$, all points on this segment are also at distance less than $\epsilon$. As $f$ is a neural network with ReLU activations, $f$ is a continuous function, as are the activation functions, $f_u$ for each of its internal neurons, $u$. Furthermore, input points at the boundary between two activation regions (i.e., $z$, such that $f_u(z) = 0$) belong to *both* activation regions. Therefore, listing all activation regions on the segment between $x$ and $p$ yields a path, $P$, from $\bar{A}_x$ to $\bar{A}_p$, with each region on the path containing an input point, $x_i$, on the segment, such that $\|x - x_i\| \leq \epsilon$.

That each $\bar{A}_i, \bar{A}_j$ in $P$ is unique follows from the convexity of activation regions: If there exist $\bar{A}_i = \bar{A}_j$ in $P$ such that there exists an $\bar{A}_k$ in between, then $\bar{A}_i = \bar{A}_j$ cannot be convex, as there exists a line segment with its end points in $\bar{A}_i$ and $\bar{A}_j$ that is not entirely contained within the region $\bar{A}_i = \bar{A}_j$. This ensures that paths are of finite length.

**(2) Exploration by the Algorithm**    Given the existence of the path, $P$, from $\bar{A}_x$ to $\bar{A}_p$, we now prove that FGP would visit all activation regions in $P$ if it returns robust. We proceed by induction on the length of paths induced (similarly to as above) by a line segment included in the $\epsilon$-ball centered on $x$.

In the base case, if the path is of length one, then it contains only $\bar{A}_x$, and the claim holds trivially since FGP starts by visiting $\bar{A}_x$.

In the inductive case, let us assume that for any path with length at most $k$ induced by a segment, $s$, beginning at $x$ with $\|s\| \leq \epsilon$, FGP visits all regions in the path. Now consider a path, $P' = \bar{A}_0, \ldots, \bar{A}_k$, of length $k+1$, induced by a segment, $s'$, beginning at $x$ with $\|s'\| \leq \epsilon$. Since $\bar{A}_{k-1}$ is on the path, there exists a point, $x_{k-1}$, on $s'$ such that the sub-segment from $x$ to $x_{k-1}$ induces a path of length $k$; thus we can apply our induction hypothesis to conclude that FGP visits $\bar{A}_0, \ldots, \bar{A}_{k-1}$. Now, since $\bar{A}_{k-1}$ and $\bar{A}_k$ are neighbors in $P'$, they must share some boundary, $C$, that is intersected by $s'$. Thus, since $\|s'\| \leq \epsilon$, $d(x, C) \leq \epsilon$; thus when FGP visits $\bar{A}_{k-1}$, it will add $\bar{A}_k$ to the queue via $C$. Therefore, since FGP returns robust only when all regions in the queue have been visited, FGP will visit $\bar{A}_k$, concluding the proof of (2).

**(3) Detection of Adversarial Examples**   We conclude by proving that if there exists an adversarial example in a region visited by FGP, then we either return not_robust or unknown.

If $p$ in $\bar{A}_p$ is an adversarial example, then $F(x) \neq F(p)$. By continuity of $f$, this means that there exists an input, $p'$ on the segment from $x$ to $p$ that is exactly on a decision boundary, $C$. As $\|x - p'\| \leq \|x - p\| \leq \epsilon$, $C$ must have been analyzed when exploring $\bar{A}_p$.

However, when analyzing a decision boundary, FGP always returns either not_robust or unknown.

Thus, the decision boundary in the activation region containing $p$ would have been analyzed by FGP; this yields a contradiction, as the algorithm must have returned not_robust or unknown rather than robust. □

## A.2   Proof of Loop Invariant 1

The proof of lower-bound variant is similar to the proof of FGP in Section A.1. It relies on the following loop invariant.

*Proof.* Loop Invariant 1 trivially holds on entry of the loop since $bound$ is initialized to 0. Proving that the invariant is maintained is more interesting. Suppose that $bound$ increases to $bound'$. It must then be shown that there is no unvisited activation region at distance less than $bound'$. We proceed again by contradiction: assume there was such a region, $\bar{A}$, containing a point, $p$, such that $\|x - p\| \leq bound'$. Again, let us consider the segment from $x$ to $p$, and the path, $P$, it induces. Let us consider $\bar{A}_i$, the first region of $P$ at distance greater than $bound$ that was not previously visited. If no such region exists, then $\bar{A}$ is at distance less than $bound$ from $x$, and so by our induction hypothesis, it was already visited. otherwise, $\bar{A}_{i-1}$ was visited, and the activation constraint, $C$, between $\bar{A}_{i-1}$ and $\bar{A}_i$ is such that $d(x, C) \leq bound' \leq \epsilon$. Therefore, $C$ (which leads to $\bar{A}_i$) was already added to the queue with priority less than $bound'$, and by virtue of the priority queue, it was explored before the current iteration, which yields a contradiction.

If $bound$ does not increase, the invariant still trivially holds. This case can happen because our computation of the distance to constraints is an underapproximation of the true distance to the feasible portion of the constraint. □

## B   Computing Activation Constraints And Decision Boundaries

Recall that an activation constraint, $C_u$, which is satisfied when $f_u(x) \geq 0 \iff A(u)$, is a linear constraint with coefficients $w_u$, and intercepts, $b_u$. The computation of these weights and intercepts does not depend on a particular point in $\bar{A}$ — only on the activation pattern, $A$. Thus, *we can compute the boundaries of an activation region, $\bar{A}$, knowing only the corresponding activation pattern, $A$.*

In practice, the coefficients, $w_u$, correspond to the gradient of $f_u$ with respect to its inputs, evaluated at a point in $\bar{A}$. However, frameworks that perform automatic differentiation typically require a concrete point for evaluation of the gradient. Thus, we compute the gradient via backpropagation with the activation vectors, $a_i$, where the position in $a_i$ corresponding to neuron, $u$, takes value 1 if $A(u)$ is $true$ and 0 otherwise. The intercepts, $b_u$, are computed via a forward computation using the activation vectors, with $x$ set to 0. These operations can easily be implemented to run efficiently on a GPU.

Decision boundaries in an activation region, $\bar{A}$, are computed similarly to internal activation constraints; we take the linearization of the network in $\bar{A}$. Since the network is piecewise linear, and linear within each activation region, this linearization is exact for the region being considered.

## C  HYPERPARAMETERS

Here, we provide details on the hyperparameters used to train the models used in our evaluation and to conduct the PGD (Madry et al., 2018) attacks used to obtain the upper bounds on the verified robust accuracy (VRA).

**CNN Architecture.**  We used a CNN architecture that has been used for benchmarking verification by (Wong & Kolter, 2018) and (Croce et al., 2019). It contains 2 convolutional layers, each using $4 \times 4$ filters with a $2 \times 2$ stride, with 16 and 32 channels respectively, followed by a dense layer of 100 nodes.

**PGD Adversarially-trained Models.**  To train models for robustness, we used PGD adversarial training (Madry et al., 2018). For training, we used the $\ell_2$ norm, and let $\epsilon = 2.5$—10 times the $\epsilon$ we verify with in order to have a higher fraction of verifiably-robust points. We trained each model for 20 epochs with a batch size of 128 and 50 PGD steps.

**Maximum-Margin-Regularization-trained Models.**  To train models for both robustness and verifiability, we used maximum margin regularization (Croce et al., 2019) (MMR). MMR has several hyperparameters including $\gamma_B$ and $\gamma_D$, which correspond to the desired distance of the training points from the internal boundaries and decision boundaries respectively (these can be seen as acting similarly to the choice of $\epsilon$ in PGD training); $n_B$ and $n_D$, which specify the number of internal and decision boundaries to be moved in each update; and $\lambda$, which specifies the relative weight of the regularization as opposed to the regular loss. See (Croce et al., 2019) for more details on these hyperparameters. We set $\gamma_B = \gamma_D = 2.5$, $n_B = 100$, $n_D = 9$, and $\lambda = 0.5$, and trained each model for 20 epochs with a batch size of 128.

**ReLU-Stability-trained Models.**  We also used ReLU stability (RS) (Xiao et al., 2019) to regularize models for verifiable robustness. We trained using RS with PGD adversarial loss and $\ell_1$ weight regularization, as was done by Xiao et al.. We weighted the RS loss by $\alpha = 2.0$, using an $\epsilon$ (i.e., the distance over which the ReLU activations should remain stable) of $8/255$, and weighted the PGD adversarial loss by $\beta = 1.0$, using an $\epsilon$ (i.e., the target robustness radius) of $36/255$. We scheduled the $\ell_1$ regularization to decay from $10^{-2}$ to $10^{-3}$ over the course of training, and trained for 100 epochs with a batch size of 128.

**PGD Attacks.**  We obtained an upper bound on the VRA by performing PGD attacks on all the correctly-classified points for which every method either timed out or reported unknown; the upper bound is the best VRA plus the fraction of points that were correctly-classified, undecided, and for which the PGD attack did not successfully find an adversarial example. We conducted these attacks with an $\ell_2$ bound of $\epsilon = 0.25$ with 1,000 PGD steps.

**Arbitrary Test Instances**  To arbitrarily pick the 100 test instances, we used the random.randint function from Numpy, setting the initial seed to 4444. The indices of the test instances are 8361, 7471, 8091, 4275, 5205, 6886, 5341, 4026, 4554, 5835, 8181, 6124, 6309, 1875, 7910, 6018, 1985, 6912, 8676, 2376, 9143, 5527, 5540, 4661, 1910, 7498, 2190, 789, 1672, 3393, 1925, 841, 8333, 4644, 5648, 138, 7872, 5204, 441, 3951, 3812, 3983, 8598, 8334, 5666, 5, 1014, 9148, 5993, 2182, 4141, 5385, 2774, 9927, 7507, 9097, 7047, 7319, 1638, 9535, 8889, 1196, 4992, 9223, 6525, 9577, 2266, 1748, 6462, 2969, 4866, 271, 9890, 2814, 9848, 8513, 289, 9002, 1103, 506, 1576, 1035, 9808, 8059, 2424, 6072, 6577, 1463, 5573, 8995, 100, 1457, 8103, 7835, 2491, 9481, 8053, 2478, 8132, 7033.

# D    BREAKDOWN OF RUN-TIMES FOR DECIDED POINTS

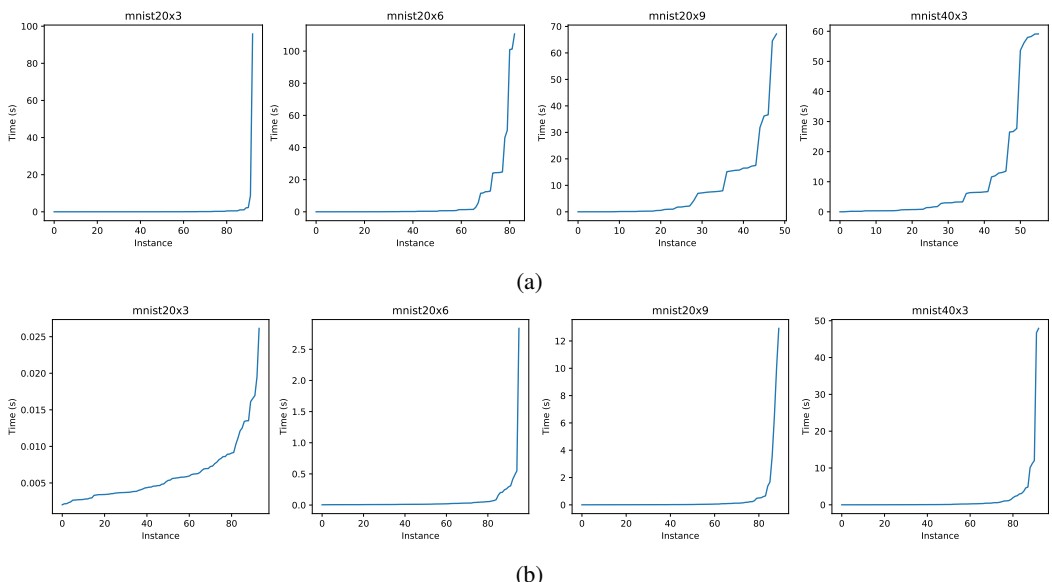

(a)

(b)

Figure 2: Cactus plots showing the time taken for each instance that is decided as either robust or not_robust. The instances are ordered by increasing time-per-instance. Results are shown for the PGD adversarially-trained models for both the $\ell_2$ norm (a) and the $\ell_\infty$ norm (b).

