# OpenReview forum: "Fast Geometric Projections for Local Robustness Certification"
_ICLR.cc/2021/Conference — ICLR 2021 Spotlight_

### Official Review · AnonReviewer3 · 2020-10-24
**General method for certifying l_p robustness**

**Rating:** 7
**Confidence:** 3

**Review:**

The authors propose a systematic search over the convex polyhedral regions on which the network is linear, to find the decision boundary, so to certify local $\ell_p$ robustness. The method is fairly general and evaluated for $p = 2$ and $p = \infty$. A key point that makes the method feasible is leveraging of the compositional structure.

The proposed verification method is incomplete and returns, given an input either one of 2 certificates (robust or not_robust) or abstains from certification. In the case of $\ell_2$ robustness, significant speed ups are gained compared to prior work. Certifying local $\ell_p$ robustness is in general an important problem. The scalability to large networks seems to be an issue, although the proposed method significantly outperforms prior work.

Suggestions:
A potential improvement for Section 2 would be to include a simple running example, that showcases the algorithm step by step. Ideally, the same example would then also be used for Figure 2.

Questions:
- What would it take to make the proposed method complete? What would the complexity / runtime be?
- How large can the networks be grown for a more generous time budget (i.e. 5 min or 10 min instead of 2 min) so that the proposed method still performs well?

Given the generality of the approach and the speed up gained compared to prior work, i give this paper an accept.

===

I thank the authors for providing the answers to my questions. I will not change my score.

---

> ### Author Response · Authors · 2020-11-22
> **Response to AnonReviewer3**
>
> We thank the reviewer for their comments on our paper. We submitted a revised version, and answer below to the different questions from the reviewer.
>
> > What would it take to make the proposed method complete? What would the complexity / runtime be?
>
> The incompleteness of our method comes from our computation of the distance between a point and an activation constraint or decision boundary. These constraints are defined in polytopes corresponding to activation regions; as such they are hyperplanes *constrained by the polytope boundaries*. We compute an underapproximation of the distance by considering the geometric distance from a point to the unconstrained hyperplanes. This is empirically more efficient, but leads to incompleteness. Several existing methods such as GeoCert or MIP compute these distances exactly using expensive QP solvers. As our evaluation shows, this leads to drastically larger runtimes. A hybrid approach for achieving completeness may be possible, though, by solving a QP only when our algorithm encounters an “unknown” result. This would be interesting to implement, although we note that on most of the points that our algorithm returns unknown, GeoCert and MIP time out, so it isn’t clear whether even this hybrid approach would be likely to terminate in a reasonable amount of time.
>
> The complexity of our algorithm is in the worst case exponential, though a “typical” complexity is hard to characterize. At a high level, the complexity will be a function of the number of regions searched and the time required to search each region. We found that while FGP may search more regions than, e.g., GeoCert (since it may search some regions that are infeasible or outside the epsilon ball), it more than makes up for this by spending less than one thousandth of the time in each region. I.e., for high-dimensional networks, solving a QP can be thousands of times slower than taking a projection. If too many QPs are required, this will significantly hurt the runtime of the algorithm.
>
>
> > How large can the networks be grown for a more generous time budget (i.e. 5 min or 10 min instead of 2 min) so that the proposed method still performs well?
>
> Any exact (or approximate) verification method that performs a search through the regions of the polyhedral complex defined by the network may need to search an exponential number of regions. While it is hard to characterize the “typical” geometry of networks obtained via training, we find that in practice, such algorithms seem to indeed have exponential runtime. Given the exponential nature of the problem being solved, as the networks become larger, the runtime increases significantly. We therefore hypothesize that in most cases, adding a more generous time budget will not enable scaling to significantly larger networks (for any of the methods evaluated). This would be an interesting experiment to run, though.
>
> On the other hand, networks that are trained for verifiability are less likely to have a “worst-case” geometry, giving more hope that a larger time budget could help.

---

### Official Review · AnonReviewer1 · 2020-10-27
**Interesting method which could benefit from further experiments**

**Rating:** 6
**Confidence:** 4

**Review:**

The paper proposes Fast Geometric Projections (FGP), a method to certify the robustness of neural networks with ReLU as activation exploiting the fact the such networks are piecewise affine functions. FGP can verify whether in an $\ell_p$-ball around an input $x$ adversarial examples exist or not. Since it's not guaranteed to find a conclusive solution, it has the option of returning "unknown" if the given point could not be certified. FGP aims at computationally efficiency at the price of incompleteness. Finally, it can provide lower bounds on the norm of the smallest adversarial perturbation.

Pros
1. Verifying efficiently the robustness of (large) neural networks is an interesting task and most of the research has primarily focused on the $\ell_\infty$-threat model. A fast methods which is effective in particular wrt the $\ell_2$-norm is valuable.
2. The proposed methods is geometrically justified and benefits from many heuristics to optimize efficiency.
3. In the experimental evaluation, for the $\ell_2$-threat model, FGP outperforms the methods for exact verification (GeoCert, MIP) with a timeout in terms of both verified robust accuracy and runtime (with a large margin). Moreover, it scales better than other methods to larger and deeper architectures when tested  on models trained for verifiability (MMR, RS).
4. When comparing lower bounds on the size of the minimal adversarial perturbations, it provides better bounds than FastLin, although with higher computational cost.

Cons
1. To achieve good provable robustness, it is necessary to train specifically for it and then usually the best bounds are given by the same technique deployed at training time (e.g. IBP training). An evaluation of FGP in such scenario seems to me particularly meaningful (and at the moment missing). If the proposed FGP outperformed the bounds given by the technique of, for example, (Wong & Kolter, 2018) on a model trained as in (Wong & Kolter, 2018), this would strengthen the proposed method.
2. The $\ell_\infty$ version doesn't seem as competitive the $\ell_2$ one. Moreover, in that case $\epsilon=0.01$ is used, which is very small for MNIST or F-MNIST (e.g. Zhang et al. (2019) have guaranteed up to $\epsilon=0.4$).
3. It seems that CROWN (Zhang et al., 2018) slightly improves upon FastLin, so it might make sense to include it in the comparison.

Other comments
1. Only one value for $\epsilon$ is tested for each dataset. It would be interesting to see how FGP scales to larger values.
2. The clean accuracy of the models used in the evaluation should be reported (it would be helpful to know whether they achieve reasonable performance).
3. Also, for completeness, using the different types of models (naturally trained, MMR and RS) for all the datasets (and also for the comparison to FastLin) would be good.

Overall, the method is clearly presented and achieves good empirical results. As mentioned above, further experiments could show its effectiveness and applicability on models with state-of-the-art provable robustness.

Zhang et al., "Towards Stable and Efficient Training of Verifiably Robust Neural Networks", 2019

---
Update after rebuttal

I thank the authors for their response. After reading it, the revised version and the other reviews, I think that there's evidence of the effectiveness of the proposed method.

As I tried to convey in the initial review, I consider the main weakness of the paper not showing the performance on models achieving state-of-the-art verified robustness. I think the case of $\ell_\infty$ is emblematic, where $\epsilon=0.01$ is used. Since there exist models achieving provable robustness >90% for $\epsilon=0.3$ (on MNIST), the scenario considered in the evaluation doesn't seem interesting, regardless of how FGP compares to the other methods.

While I agree with the authors that training for provable guarantees sacrifices clean accuracy in most of the cases, as far as I know that is currently the way to achieve good VRA at meaningful thresholds. Along this line, also the authors used FGP on larger models when those were trained with MMR or RS. In my opinion the most interesting application of a (incomplete) verification method like FGP is to provide better VRA than current methods for training (IBP, CROWN-IBP, Wong & Kolter's method), for example using less heavily regularized models which can retain higher clean accuracy. I think this is the missing piece in the current version of the paper.

For these reasons, I keep my initial score.

---

> ### Author Response · Authors · 2020-11-22
> **Response to AnonReviewer1**
>
> We thank the reviewer for their comments on our paper. We submitted a revised version, and answer below to the different questions from the reviewer.
>
> > 1. To achieve good provable robustness, it is necessary to train specifically for it and then usually the best bounds are given by the same technique deployed at training time (e.g. IBP training).
>
> The advantage to our approach is that it does not require a particular training routine. Several of the models we evaluated on were not trained for verifiability at all. Nonetheless, we achieve within a few percentage points of the upper bound on the VRA on several of these models. On the other hand, training for provable robustness may impose a significant cost on the model’s accuracy. For example, the CIFAR10 model trained with RS achieved only 27% VRA. Perhaps the largest verifiable models, ResNet models trained with DiffAI achieve as low as 20% VRA (lower than would be achieved by a linear model). While some of the current best benchmark VRAs today do in fact result from training for provable robustness and verifying robustness with a method tailored to the training technique, we suggest that it may not be the case that this will achieve the best trade-off between robustness and accuracy in the future.
>
> > 2. The l_inf norm version doesn't seem as competitive as the l_2 norm one.
>
> Our algorithm is indeed more effective when using the l_2 norm than when using the l_inf norm.
> As stated in Section 3.3 in the original version of the paper, there are several explanations for this. First, most existing tools rely on constraint solving, which is more efficient when using the l_inf norm as the objective function is linear instead of quadratic, which explains the smaller performance gain when comparing speed of the analyses. Second, a geometric projection in the l_2 space is unique, while it is not always the case when working with the l_inf norm. In the latter case, we pick an arbitrary point as the projection, which seems to rarely be an adversarial example. This explains the higher rate of unknown results.
>
> We revised the paper to include the detailed evaluation of l_inf results in Section 3 instead of the appendix. It is worth noting that although less precise, our approach is nevertheless faster than existing tools. Furthermore, we argue that l_2 is also an important norm to consider for local robustness; and it has been less-well studied and proved more difficult.

---

### Official Review · AnonReviewer4 · 2020-10-28
**Novel idea for accelerating verification**

**Rating:** 8
**Confidence:** 5

**Review:**

Title before Rebuttal: _Novel idea for accelerating verification hampered by concerns on correctness of algorithm_

# Summary of Contributions
The paper proposes a method to verify local robustness networks with piecewise-linear activation functions, providing tighter bounds than existing approximate verification approaches while scaling to deeper networks than existing complete verifiers.

Furthermore, unlike other approximate verification techniques, the method is able to show the existence of adversarial examples (in some cases).

The method works by exhaustively checking each of the activation regions (that is, the convex polytopes on which the network is linear) that are fully or partially within the ε-ball for the presence of a decision boundary.
- If there exists a region containing a decision boundary for which the projection $p$ of $x$ to the decision boundary is within the ε-ball:
  - An adversarial example exists if $p$ is in the activation region
  - The robustness status is unknown if $p$ lies outside of the activation region
- Otherwise, if no such region exists, the sample is robust to local perturbations.

# Score Recommendation
The method in this paper is novel and the results presented are compelling. However, it is currently marginally below the acceptance threshold (5) for me, as I am concerned about the correctness of the optimized tree-based exploration variant of FGP presented. Notwithstanding the other issues identified, if my concerns around the correctness are addressed and the presented results stand, this paper would be among the top 50% of accepted papers (8). Otherwise, it would be a reject (3).

For the tree-based exploration approach to be correct, we need to prove that regions filtered out are unreachable or explored through a different path. The proof in A.2 needs to be expanded upon, and does not currently provide sufficient detail to demonstrate that this is the case.

Specifically, consider the potential counterexample in the image linked: https://www.geogebra.org/geometry/ajzcrves. We have four polytopes A_prev, A, A_next, A bounded by ECG, DCE, FCD, GCF respectively. Every region that differs by one neuron is adjacent to each other. A_next is discarded when exploring from A, it will not be reached by any other path since A' will not be explored from A_prev (the boundary GC does not pass inside the ε-ball). I expect that some other property of the network means that this is not an actual counterexample, but do not see which specifically.

# Strengths
*Novelty*: The paper presents a novel approach for verification that can be accelerated on GPUs. While other work to accelerate verification on GPUs exist (e.g. [predicting dual variables for optimization using a neural network](https://arxiv.org/pdf/1805.10265.pdf) or [expressing SDP solver algorithms as passes through the network being verified](https://arxiv.org/pdf/2010.11645.pdf)), the method presented simply exhaustively searches (very efficiently) each activation region.

*Results*: To the best of my knowledge, the method:
- Advances the state-of-the-art in certifying local stability over perturbations of bounded $l_2$-norm, with significant improvements in runtime (1-2 orders of magnitude) over GeoCert.
- Advances the state-of-the-art in providing tight certified lower bounds (up to 2 orders of magnitude) on $l_2$-robustness radius

*Overall Clarity*: This paper was a pleasure to read, and I wanted to make special note of this. I appreciated that thought had been put in in naming and defining terms (e.g. activation pattern, activation region), and found the diagrams in particular helped me understand the algorithm. Results were organized clearly

*Reproducibility*: The authors took pains to provide details that make their results more reproducible (architecture, training hyperparameters and method, time budget, computational setup for verification).

# Weaknesses
_(The relative length of this section should not be taken as an indictment of this paper.)_

In addition to the issue with the optimized variant of FGP, here are some additional weaknesses that should be addressed.

## Clarity of Algorithm Description
- With reference to Figure 2b and focusing on $C_1$, the penultimate paragraph of page 3 says “For each constraint at distance less than ε from x, we enqueue the corresponding activation region if it has not been searched yet”. It would be helpful to emphasize in the paper that we are checking if the *line* $C_1$ intersects the ε-ball, not just the line _segment_ between $A_{11}$ and $A_{01}$.
- I could not find an explanation in the paper for how the decision boundaries are computed; this seems core to the efficiency of the approach. As far as I can tell, the method described linearizes the network $f$ about the activation region (obtaining $f’$), and computes the decision boundary in $f’$ (not $f$), and then checks the distance to $x$. If this is the case, it should be explicitly specified, and a concise proof that this is correct added.

## Generalization to Other Norms
- The time comparison (“up to 5x faster than ERAN on models trained with PGD, and one to two orders of magnitude faster than ERAN on models trained using MMR”) is misleading without noting that the FGP returns “unknown” for a significant proportion of samples (6-15%) while ERAN+MIP returns a result in all cases. This should be noted in Section 3.3, rather than readers having to head to the appendix to determine this.
- While ERAN is a reasonably competitive verification method, I’d be interested to see a comparison with nnenum (CAV ‘20: https://github.com/stanleybak/nnenum) if possible, as I expect that nnenum may provide stronger results than ERAN.
  - I would also love to see a comparison with [VeriNet](https://vas.doc.ic.ac.uk/software/neural/) (ECAI ‘20 but this is not necessary as ECAI ‘20 did occur after Aug 2, 2020.

## Certified Lower Bounds
- Given that FastLin is designed to be quick but provide loose bounds, it would be reasonable to provide the mean runtime for FastLin (presumably ~1s) and FGP_LB (60s) in Table 1b, so that someone scanning the paper can understand that FGP provides better bounds at the cost of longer runtime)
- CROWN provides better certified lower bounds than FastLin without a significantly higher cost to runtime (see, for example, Table 4 of the original paper). Computing the mean bounds in Table 1b using CROWN (rather than FastLin) would be a better comparison of FGP to the state of the art.

## Reproducibility
- The first paragraph of page 6 states that “measurements are obtained by evaluating on 100 arbitrary instances”. It would be helpful for reproducibility to provide the indexes of these instances.
- It would be nice to have the relevant code released.

# Questions for Authors
- With reference to Figure 1b: it seems possible to distinguish between the left and right cases by computing the projection of point $p$ onto the ε-ball, $p’$, and checking whether $p’$ is within the activation region. This would allow the algorithm to avoid returning `unknown` in cases like the right. Is this incorrect, or was it merely challenging to implement?
- How did you handle the presence of multiple decision boundaries within a single activation region (e.g. one between category 1 and 2, and one between category 1 and 3) or does this provably not occur?

# Additional Feedback

- Figure 1 combines two related but different ideas (an illustration of the basic FGP algorithm, and an example of a case requiring FGP to return unknown when analyzing a boundary constraint) into a single figure. As a result, the figure was slightly confusing for me when I initially looked at it. I understand that this is probably done for space constraints, and the authors do attempt to use the spacing between the squares to distinguish between the two groups of squares, but perhaps a light vertical line (or some other visual aid) would help to distinguish more clearly. (I have a similar recommendation for Figure 2 is even less clear).
- The use of “enriched” in third paragraph of Page 5 is slightly confusing (is the queue somehow enriched, or is it the regions that are?). What about “Instead, it is sufficient to use a regular queue of regions, augmenting _each region_ with a field storing the layer the last flipped neuron belongs to”?
- The paragraph on “depth scalability” discussing Figure 3 seems to emphasize the _ratio_ of regions searched between FGP and GeoCert. Given this, showing the y-axis of regions searched on a log scale might be appropriate since it enables readers to do the comparison themselves.
- The layout of the paper means that the first introduction in text of verified robust accuracy occurs after its use in Table 1. Given that it looks like there is enough space in the caption, it might be worth it to spell out what VRA is in the caption.
- For reproducibility, it would be helpful to indicate which commit / [release](https://github.com/vtjeng/MIPVerify.jl/releases) of the MIP code you used to obtain the results.

# Post-Rebuttal Comments
I've increased the rating for the paper from 5 -> 8 as the authors have addressed all my substantive concerns.

- Most critically, the authors demonstrated that, even without the tree-based exploration variant, FGP's performance improves significantly over the state of the art particularly for $l_2$ networks.
- The authors have also significantly improved the clarity of the algorithm description, made the comparison in the section on generalization to other norms more clear, provided information in the paper allowing readers to understand that FastLin is significantly faster (but also significantly looser), and provided details that make reproducing these results far simpler.

---

> ### Author Response · Authors · 2020-11-22
> **Response to AnonReviewer4**
>
> We thank the reviewer for their comments on our paper. We submitted a revised version, and answer below to the different concerns raised by the reviewer.
>
> > Correctness of the tree-based heuristic
>
> We thank the reviewer for bringing concerns about the tree heuristic to our attention. While the counterexample suggested in the review is not actually realizable by a neural network (e.g., when two boundaries intersect, at most one of them can bend), it led us to discover the following similar counterexample that is realizable by a feed-forward network (https://www.geogebra.org/geometry/dwtt8p9x). Critically, the correctness proof for this optimization was short on detail; while it correctly states that A_10 would be visited, it failed to consider the cases in which A_11 will not be explored from A_10 (as shown in the counterexample). After further inspection and review, the rest of the proofs of correctness continue to hold.  We have revised our paper to remove the tree-based exploration heuristic and report on the results of re-running all of our experiments without that heuristic. Our new experimental results show that this optimization only had a small impact on the median verification time -- we still outperform existing tools by several orders of magnitude. The biggest difference is in the number of test instances that time out. What this seems to suggest is that a minority of inputs especially benefited from the tree-based exploration, while the majority of the test instances did not.
>
> Empirically, it seems that cases where this heuristic led to unsoundness were very rare, as our experimental results were previously consistent with those reported by exact techniques (i.e., there were no examples of instances proved robust via our technique and not robust by another, or vice versa). Nevertheless, as shown by our revised counterexample that it is possible to artificially construct networks where counterexamples do exist. Whether existing training techniques preclude such counterexamples remains an open question.
>
> > Clarity of algorithm description
>
> Our revised paper clarifies the presentation of the algorithm in section 2.1, stating more clearly that we check whether the infinite prolongation of the activation constraint (i.e. the line) intersects the epsilon-ball.
>
> > How are decision boundaries computed?
>
> Decision boundaries are computed similarly to internal boundaries, but with a slight modification. Essentially an *internal* boundary is where a neuron flips from on to off, i.e., where the value of that neuron is equal to zero. A *decision* boundary is still linear, but for softmax models, rather than corresponding to the value for a particular class being zero, it corresponds to where the value for that class is larger than for the other classes. If the original class at point x is c, the boundary for class c’ is where f_c = f_c’.
>
> > Generalization to other norms
>
> We included the full analysis of results for the l_inf norm in Section 3.3, fully comparing how FGP fares compared to existing techniques in the paper itself.
> We thank the reviewer for raising nnenum to our attention. Unfortunately, our tool currently does not support the input format that nnenum requires.
>
> > Certified Lower Bounds
>
> We added the mean runtime for FastLin inside our evaluation section. While perhaps the primary contribution of CROWN over FastLin is its ability to generalize to more general activation functions, it is indeed tighter than FastLin. The authors report a fairly consistent 20% improvement on the lower bounds provided by CROWN as compared to FastLin. We found that on typical instances, FastLin achieved 4% to 69% of the bounds found by FGP, thus we might expect CROWN to achieve 5% to 83% of the bounds found by FGP. We will look into extending our experiments to include CROWN, however we believe the findings will be qualitatively the same: that CROWN, like FastLin, is often substantially looser than FGP.
>
> > Reproducibility
>
> We added information about how test instances were arbitrarily chosen to the appendix.  We also added the commit hash of GeoCert and the release version of MIP we used during our evaluation.  We will release our code with an open-source license upon acceptance of the paper.

---

> > ### Author Response · Authors · 2020-11-22
> > **Reply to Additional questions from AnonReviewer4**
> >
> > > With reference to Figure 1b: it seems possible to distinguish between the left and right cases by computing the projection of point p onto the ε-ball, p’ , and checking whether p’ is within the activation region. This would allow the algorithm to avoid returning unknown in cases like the right. Is this incorrect, or was it merely challenging to implement?
> >
> > Unfortunately, this heuristic does not work in higher dimensions. Figure 1b assumes that the input space has dimension 2, hence projection point p onto the epsilon-ball results in two points that can be easily checked. In higher dimensions, the intersection of the hyperplane corresponding to the decision boundary with the epsilon-ball will result in an infinite number of points.
> >
> > > How did you handle the presence of multiple decision boundaries within a single activation region (e.g. one between category 1 and 2, and one between category 1 and 3) or does this provably not occur?
> >
> > When trying to prove that an input with classification 1 for instance was locally robust, we considered *al*l decision boundaries involving class 1 within the activation region (i.e., we considered both the boundary between 1 and 2 and between 1 and 3). It might be the case that projecting on the decision boundary between category 1 and 2 actually leads to yet a different classification. For local robustness, this is irrelevant: as long as the classification is not 1, the projection is an adversarial example.

---

> > > ### Comment · AnonReviewer4 · 2020-11-23
> > > **Addressing the unknown case**
> > >
> > > Thank you for your explanation! I hadn't been thinking in higher dimensions. It would still be really nice to be able to avoid falling back to full constraint solving in this case; would the following approach work?
> > >
> > > Given an input space of dimension $d$, we would like to check whether any point in the [$d-1$-ball](https://en.wikipedia.org/wiki/Ball_(mathematics)) corresponding to the region of the boundary constraint $C_B$ that is also in the $\epsilon$-ball (which I'll call the "projection ball") intersects with the hyperplane defined by the activation constraint $C_u$. The properties of the activation constraints are already known, and the center of the "projection ball" is $p$ and the radius should be $\sqrt{\epsilon^2-d^2}$. I haven't taken a thorough look, but it looks like [this stackexchange answer](https://math.stackexchange.com/questions/1819802/intersection-of-hypersphere-and-hyperplane-question) provides a solution? (Alternatively, if an analytic solution doesn't exist, this can be expressed as a quadratically constrained convex optimization problem).

---

> > > > ### Author Response · Authors · 2020-11-24
> > > > **Addressing the unknown case**
> > > >
> > > > This is a very interesting idea, and it is similar to other heuristics we explored in the past.
> > > > Based on our experience, it is likely that there exists a counterexample to this approach, although none immediately comes to mind.
> > > > We would need to investigate more to try to prove or disprove the correctness of such a fall back method

---

> > > > > ### Comment · AnonReviewer4 · 2020-11-24
> > > > > **Addressing the unknown case**
> > > > >
> > > > > Thank you - I'd be interested to get updates if you can prove this or find a counterexample!

---

> > ### Comment · AnonReviewer4 · 2020-11-23
> > **Reply to Author Response**
> >
> > Thank you for your response and for updating the paper.
> >
> > Overall, I'm really glad to see that the algorithm still has good performance even without the tree-based heuristic. As promised, I will be updating my score recommendation on the paper.
> >
> > I do have a few additional follow-ups:
> >
> > ## Clarity of Algorithm Description
> >
> > I was initially confused by this section of the explanation:
> >
> > > ... we take the projection, $p$, from $x$ onto $C$ ... if $p$ has the same class as $x$, ...
> >
> > The reason why I was confused is that I expected that since $p$ lay on a decision boundary, it would be _by definition_ an adversarial example that could not be in the same class as $x$. What I had missed was that $p$ was the projection of $x$ onto the infinite prolongation of the activation constraint in $\bar{A}$, and that $p$ might not lie on an activation constraint if it is not in $\bar{A}$. In particular, with reference to Figure 1b), the decision boundary $C_B$ is only valid in the white region $\bar{A}$, and _not_ in the gray region. I think it's worth emphasizing this. Perhaps you could add in the figure a solid blue line showing that the decision boundary in the gray area does not have to be aligned with $C_B$?
> >
> > ## How are the decision boundaries computed?
> >
> > I understand that the decision boundary is linear _within a particular activation region_, and that it corresponds to the value for the class being equal to another class. My uncertainty is around how you are computing this line segment. Is it the case that you are linearizing the network $f$ about the activation region (obtaining $f'$), and computing the decision boundary in $f'$ (and not $f$), with the understanding that the decision boundary _in the activation region_ is the same between $f'$ and $f$? If so, I think it's worth spelling it out; otherwise, I'd like to understand how you compute the decision boundary.
> >
> > ## Generalization to other norms
> >
> > The comparison with ERAN+MIP is still slightly complicated since ERAN+MIP returns a definite answer (either R or NR) on every sample, while FGP does not, with U or TO between 5-15% of the time. For example, for `fmnist100x6*`, FGP is only able to provide an answer for 92 samples. It is possible that the average time for ERAN+MIP on the fastest 92 samples is _lower_ than FGP, with the remaining 8 samples bringing up the average amount of time. A cactus plot in the appendix showing the amount of samples where a definite answer is returned against time (see Fig. 1 of [this paper](http://stanleybak.com/papers/bak2020cav.pdf) for an example) for each of the networks would help readers answer this question for themselves.
> >
> > Also, it's unfortunate that you were not able to do the conversion to the format required by `nnenum`, but I understand that that was a big ask in the short period of time available.
> >
> > ## Reproducibility
> >
> > It would be really helpful for future authors comparing their work to this paper (or others attempting to reproduce your results) to be able to do a sanity check with the precise subset of samples you selected. As such, I think it's worth providing exact details in the paper (rather than just in the released code, since that can be harder to find). Would it be right to say that you called `numpy.random.seed(seed=4444)`, followed by `numpy.random.randint(0, 9999, 100)`, yielding the following sequence?
> >
> > ```
> > array([8361, 7471, 8091, 4275, 5205, 6886, 5341, 4026, 4554, 5835, 8181,
> >        6124, 6309, 1875, 7910, 6018, 1985, 6912, 8676, 2376, 9143, 5527,
> >        5540, 4661, 1910, 7498, 2190,  789, 1672, 3393, 1925,  841, 8333,
> >        4644, 5648,  138, 7872, 5204,  441, 3951, 3812, 3983, 8598, 8334,
> >        5666,    5, 1014, 9148, 5993, 2182, 4141, 5385, 2774, 9927, 7507,
> >        9097, 7047, 7319, 1638, 9535, 8889, 1196, 4992, 9223, 6525, 9577,
> >        2266, 1748, 6462, 2969, 4866,  271, 9890, 2814, 9848, 8513,  289,
> >        9002, 1103,  506, 1576, 1035, 9808, 8059, 2424, 6072, 6577, 1463,
> >        5573, 8995,  100, 1457, 8103, 7835, 2491, 9481, 8053, 2478, 8132,
> >        7033])
> > ```

---

> > > ### Author Response · Authors · 2020-11-24
> > > **Additional response**
> > >
> > > We thank the reviewer for the additional feedback. Below are answers to the different questions
> > >
> > > > Clarity of Algorithm Description
> > >
> > > Adding a solid line in the grey region corresponding to the true activation boundary is a great idea. We will update the figure and caption accordingly
> > >
> > > > Computing Decision Boundaries
> > >
> > > This is correct: we compute the decision boundary in region A by taking the linearization of the network in A. Note that f is piecewise linear, and linear within each activation region, thus, this linearization is exact for the region being considered.
> > >
> > > Essentially, given “activation matrices” A1, A2, …, for each layer, which are diagonal matrices with a AL_{i,i} = 1 if neuron i in layer L is active and 0 otherwise, the network can be written as
> > >
> > > f(x) = ((x W1 + b1) A1 W2 + b2) …
> > >
> > > which is a linear function of x for fixed A1, A2, …. This is essentially how we compute both the internal and decision boundaries -- it can be done essentially with a single forward pass of the network.
> > >
> > > > Generalization to Other Norms
> > >
> > > We are currently working on a cactus plot as suggested by the reviewer to provide more information about our comparison to ERAN+MIP, and will update a revised version shortly.
> > > It is worth noting that, in our evaluation section, we compare the median execution times, not the averages. As such, it is unlikely that the small number of unknown results would have a sufficient impact to change the trends of the evaluation.
> > > Furthermore, an execution of FGP leading to an unknown result is usually slower than executions leading to robust or not robust results. This is due to the heuristic presented in Section 2.2 of the latest revision of the paper, called “Exploring the Full Queue”, which helps reduce the number of unknown results reported, at the cost of a small increase in computational effort.
> > >
> > > > Reproducibility
> > >
> > > The code presented by the reviewer corresponds to what we did in our experimental setup.
> > > We nevertheless added the exact list of indices to the appendix in our latest revision

---

> > > > ### Comment · AnonReviewer4 · 2020-11-24
> > > > **Response to additional response**
> > > >
> > > > *Reproducibility*: Thank you for adding the details in the appendix!
> > > >
> > > > *Computing Decision Boundaries*: If you have the time, it would be useful to include the description above on computing decision boundaries in the paper (at least in the Appendix) for the avoidance of doubt. (Sorry if you've already done so in the most recent update.)
> > > >
> > > > *Generalization to Other Norms*: I missed that you were presenting median rather than average times; sorry. A cactus plot would still be a valuable addition if it's not too much work - I appreciate you taking the time to add it!

---

> > > > > ### Author Response · Authors · 2020-11-25
> > > > > **Additional response**
> > > > >
> > > > > We uploaded a new revision of our paper addressing your most recent comments.
> > > > >
> > > > > _Computing Decision Boundaries_: We added some additional discussion in appendix B to explain how we are computing decision boundaries.
> > > > >
> > > > > _Clarity of Algorithm Description_: We updated Figure 1(b) to present unknown cases with a solid line in the grey area.
> > > > >
> > > > > _Generalization to Other Norms_: We added cactus plots to the appendix presenting the number of robust, not robust, and unknown decisions over time when using FGP, which gives more insight into how unknown decisions impact the reported median verification time.
> > > > > When evaluating ERAN+MIP, we unfortunately only stored the aggregate of results, and thus we will need to rerun those experiments so that we can provide a similar cactus plot for ERAN in our next revision. As discussed above, our use of the median to aggregate results suggests that the handful of unknown results are not unduly skewing the comparison.
> > > > >
> > > > > We want to thank you for your continued interest in our work.  We hope that we have addressed all your concerns throughout the review process. Your suggestions have significantly increased the quality of our work and of our presentation.

---

> > > > > > ### Comment · AnonReviewer4 · 2020-11-25
> > > > > > **Thank you!**
> > > > > >
> > > > > > Thank you for your responsiveness during the review process; you have addressed all of my concerns through the review process. As it stands, this paper would be among the top 50% of accepted papers. I'm glad that the suggestions have been helpful in improving the work.
> > > > > >
> > > > > > A note about the cactus plots: they typically show times _per sample_ (not total time across all the samples), and are sorted in increasing order of time (whether the sample was determined to be robust or not robust, but excluding samples which are unknown).

---

### Official Review · AnonReviewer2 · 2020-10-29
**Interesting algorithm, but the paper could be much better by situating itself better compared to existing work.**

**Rating:** 7
**Confidence:** 4

**Review:**

Summary:
This paper proposes an algorithm to verify whether or not there exists an adversarial example in an Lp ball of size espilon around a given training sample. As opposed to the more common used bound propagation method (Crown, fastlin, IBP...), it does not do so by overapproximating bounds on the output, but instead by exploring neighbouring activation patterns to the one where the sample to verify lie, in a form of exhaustive search. As opposed to previous similar algorithm (Geocert), it sacrifices completeness in order to gain efficiency.
In addition, the authors explain how to take advantage of the Neural network structure to make the algorithm more efficient. Experimental results are very convincing, showing strong improvements in term of runtimes of the method with the proportions of "unknown" sample remaining moderate.

Main comment:
It would be beneficial for the authors to make explicit the difference between the Fast Geometric Projections algorithm and the GeoCert algorithm. As it is, it seems obvious to me that the two algorithms are clearly related but it would be beneficial to make the difference more apparent. There is a short mention in the Related Work section but I found it pretty lacking. Why does GeoCert need to rely on projection to polytope solved by quadratic program and FGP does not?
As it is, my understanding of the difference is that GeoCert, when computing the projection, computes it only to the actual existing part of the decision boundary/activation constraint, rather than to its infinite prolongation as is done by FGP. Is that correct?
Is there a intuition or a result that could be given of when GeoCert and FGP are going to return the same result? Or equivalently, a characterization of FGP incompleteness? Is it something like "There exists an activation pattern A intersecting with the epsilon ball, with the decision boundary Ca going through it, but the projection of the center point onto the (extended to an infinite line) decision boundary Ca does not belong to A"
I think that this would make the paper significantly stronger, if things were formulated in the way of "We perform this change, which leads to the loss in completeness due to not handling those cases well, but in exchange the computation that needs to be done becomes much simpler leading to X orders of magnitude improvement.", rather than "here is an algorithm that works X order of magnitude better". The first allows the reader to get an insight into the trade-off and the structure of the problem.

Smaller comment:
It would be nice to have an ablation study / experimental evaluation of the benefits brought by 2.2 and taking advantage of the network structure if it's an experiment that could easily be run.

Experiment Section comments:
- I'm unclear as to what the upper bound on the verified robust accuracy tells. For FGP on cifar-CNN, why isn't the robust accuracy 86%? It seems like all samples have been successfully handled and 86 out fof the 100 have been found robust?
- Fastlin has mostly been superseded by Crown, which usually give tighter bounds. I assume that the tightness comparison would still go the same way but it would be a more appropriate baseline. It would also be good to provide the runtime for Fastlin/Crown for the benefit of the reader. FGP takes 60s on those networks , but I doubt that Fastlin takes more than 1s. There is even a lot of space on the side of Table 1.b to add a runtime column!

Minor Notes:
- In 2.1, "if A0 contains a decision boundary" -> "contains a decision boundary C", maybe? C doesn't seem to be introduced anywhere (there is C_u for activation constraints).
- In 2.1,  at the end, it is not clear if the enqueued neighbouring activation regions will also potentially enqueue their neighbouring regions? I think that I figured out that it is based on the rest of the paper but it would be nice to make it more explicit.
- In 2.2, it would be nice, even if the proof is not there to have the intuitions behind it be in the main paper. The appendix sentence "To prove the correctness of our treebased exploration, we only need to prove that the additional regions we filter out either are unreachable, or are explored through a different path." is perfectly sufficient to convince the reader of the idea.

Recommended Related works:
The authors may potentially add a reference to "Measuring Neural Net Robustness with Constraints", by Bastani et al., NeurIPS 2016,   for context, although the method in that paper was only looking in a single linear piece of the network. Not really necessary though.


Opinion:
The algorithm is great and its presentation from a point of view of "what is done" is quite clear but the paper would in my opinion be much better if it contextualize its contribution better in comparison to other algorithms. I would be happy to raise my rating if this was the case.


###########################################
Update after discussion with the authors and reading the updated version:

The authors have clarified the points that were unclear. I'm happy to raise my score.

---

> ### Author Response · Authors · 2020-11-22
> **Response to AnonReviewer 2**
>
> We thank the reviewer for their comments on our paper. We submitted a revised version, and answer below to the different concerns raised by the reviewer
>
> > Why does GeoCert need to rely on projection to polytope solved by quadratic program and FGP does not?
>
> The reviewer’s impression is correct: while GeoCert (and MIP) compute the exact distance to an activation constraint by taking into account the boundaries of the polytope this constraint belongs to, we instead compute an underapproximation of this distance, by computing the geometric projection to the infinite prolongation of the activation constraint.
> Our updated paper clarifies this point, both in the presentation of the algorithm (Section 2.1) and in the related work.
>
> > Is there a intuition or a result that could be given of when GeoCert and FGP are going to return the same result? Or equivalently, a characterization of FGP incompleteness?
>
> Characterizing precisely when results using GeoCert and FGP diverge is tricky. Consider the example presented in Fig 1.b, where the projection p to the decision boundary is outside the activation region currently under analysis. When p is not an adversarial example, we cannot conclude and return unknown, as stated in the paper. But p could be an adversarial example, for instance if the decision boundary remains the same in the grey region, or if it “bends” towards x, in which case we would return not_robust, as GeoCert would.
>
> > For FGP on cifar-CNN, why isn't the robust accuracy 86%?
>
> Regarding the robust accuracy for cifar-CNN, robust accuracy is defined as a point being locally robust, but also correctly classified. The low robust accuracy reported suggests that, despite the majority of points being locally robust, the network itself has low accuracy. This is typical for large networks that are regularized for verifiable robustness.
>
> > Fastlin has mostly been superseded by Crown
>
> While perhaps the primary contribution of CROWN over FastLin is its ability to generalize to more general activation functions, it is indeed tighter than FastLin. The authors report a fairly consistent 20% improvement on the lower bounds provided by CROWN as compared to FastLin. We found that on typical instances, FastLin achieved 4% to 69% of the bounds found by FGP, thus we might expect CROWN to achieve 5% to 83% of the bounds found by FGP. We will look into extending our experiments to include CROWN, however we believe the findings will be qualitatively the same: that CROWN, like FastLin, is often substantially looser than FGP.

---

### Decision · Program_Chairs · 2021-01-07
**Final Decision**

**Decision:**

Accept (Spotlight)

**Comment:**

The paper presents a sound and efficient (but not complete) algorithm for verifying that a piecewise-linear neural network is constant in an Lp ball around a given point. This is a significant contribution towards practical protection from adversarial attacks with theoretical guarantees. The proposed algorithm is shown to be sound (that is, when it returns a result, that result is guaranteed to be correct) and efficient (it is easily parallelizable and can scale to large networks), but is not complete (there exist cases where the algorithm will return "I don't know"). The experiments show good results in practice. The reviewers are positive about the paper, and most initial concerns have been addressed in the rebuttal, with the paper improving as a result. Overall, this is an important contribution worth communicating to the ICLR community, so I'm happy to recommend acceptance.